# Suberoylanilide hydroxamic acid attenuates cognitive impairment in offspring caused by maternal surgery during mid-pregnancy

Yunlin Feng[1,2‡], Jia Qin[1‡], Yanfei Lu[1], Mengdie Wang[1], Shengqiang Wang[3], Foquan Luo[1]*

**1** Center for Rehabilitation Medicine, Department of Anesthesiology, Zhejiang Provincial People's Hospital, Affiliated People's Hospital, Hangzhou Medical College, Hangzhou, Zhejiang, China, **2** Department of Anesthesiology, The Affiliated Hospital of Guizhou Medical University, Guiyang, China, **3** Department of Anesthesiology, Yichun People's Hospital, Yichun, China

‡ YF and JQ contributed equally to this work as co-first authors.
* lfqxmc@ncu.edu.cn

**Data Availability Statement:** All relevant data are within the paper and its Supporting Information files.

## Abstract

Some pregnant women have to experience non-obstetric surgery during pregnancy under general anesthesia. Our previous studies showed that maternal exposure to sevoflurane, isoflurane, propofol, and ketamine causes cognitive deficits in offspring. Histone acetylation has been implicated in synaptic plasticity. Propofol is commonly used in non-obstetric procedures on pregnant women. Previous studies in our laboratory showed that maternal propofol exposure in pregnancy impairs learning and memory in offspring by disturbing histone acetylation. The present study aims to investigate whether HDAC inhibitor suberoylanilide hydroxamic acid (SAHA) could attenuate learning and memory deficits in offspring caused by maternal surgery under propofol anesthesia during mid-pregnancy. Maternal rats were exposed to propofol or underwent abdominal surgery under propofol anesthesia during middle pregnancy. The learning and memory abilities of the offspring rats were assessed using the Morris water maze (MWM) test. The protein levels of histone deacetylase 2 (HDAC2), phosphorylated cAMP response-element binding (p-CREB), brain-derived neurotrophic factor (BDNF), and phosphorylated tyrosine kinase B (p-TrkB) in the hippocampus of the offspring rats were evaluated by immunofluorescence staining and western blot. Hippocampal neuroapoptosis was detected by TUNEL staining. Our results showed that maternal propofol exposure during middle pregnancy impaired the water-maze learning and memory of the offspring rats, increased the protein level of HDAC2 and reduced the protein levels of p-CREB, BDNF and p-TrkB in the hippocampus of the offspring, and such effects were exacerbated by surgery. SAHA alleviated the cognitive dysfunction and rescued the changes in the protein levels of p-CREB, BDNF and p-TrkB induced by maternal propofol exposure alone or maternal propofol exposure plus surgery. Therefore, SAHA could be a potential and promising agent for treating the learning and memory deficits in offspring caused by maternal nonobstetric surgery under propofol anesthesia.

**Funding:** The National Natural Science Foundation of China (81960211, 81460175 and 81060093), the Natural Science Foundation of Zhejiang Province (ZCLZ24H0901) and the Zhejiang Medicine and Health Science and Technology project of Zhejiang Provincial Health Commission (2023KY037). The funders had no role in study design, data collection and analysis, decision to publish, or preparation of the manuscript.

**Competing interests:** The authors declared no competing interests.

## Introduction

Growing evidence indicates that commonly used anesthetics can cause long-term neurotoxicity in the developing brain [1–5]. Surgery may induce neurodevelopmental impairment and cognitive dysfunction in children [6]. Some pregnant women have to experience non-obstetric surgery during pregnancy under general anesthesia [7]. Brain development starts with the formation of the neural tube at week 3 in humans, that is, in the first month of the first trimester [8]. Previous studies have shown that maternal exposure to sevoflurane, isoflurane, propofol and ketamine induces cognitive deficits in offspring [9, 10]. In clinical practice, anesthesia is frequently performed because of surgery. However, the potential effect of non-obstetric surgery during pregnancy on cognitive functions of offspring and its underlying mechanism are still poorly understood.

Synaptic plasticity is essential for hippocampus-dependent learning and memory [11]. Histone acetylation, which is co-regulated by histone acetyltransferase (HAT) and histone deacetylase (HDAC), has been implicated in synaptic plasticity [12, 13]. Neonatal exposure to sevoflurane or isoflurane could induce abnormal histone acetylation in the hippocampus and neurocognitive impairment [14], and such effects could be alleviated by restoration of normal histone acetylation [15, 16]. HDAC inhibitors (HDACi) could improve memory in animals having experienced massive neurodegeneration [17] or post-traumatic stress disorder [18].

Suberoylanilide hydroxamic acid (SAHA), an HDAC inhibitor, was shown to attenuate sevoflurane-induced deficits in learning and memory in fetal mice [19]. HDAC2 is the major target of HDACi in eliciting memory enhancement [20], and over-expression of HDAC2 reduces the level of phosphorylated cAMP response-element binding protein (p-CREB) [21]. Propofol is commonly used in clinical practice, including non-obstetric procedures on pregnant women. Propofol is a fat-soluble intravenous anesthetic that can easily pass through the placental barrier [22]. It has been demonstrated that the level of propofol in newborn plasma at the time of delivery depends on that in maternal plasma [23]. Recent evidence shows that propofol can also cause neurotoxicity in developing brains [24, 25]. Previous studies in our laboratory showed that maternal propofol exposure in pregnancy impairs learning and memory in offspring by disturbing histone acetylation [26] and BDNF-TrkB [27] in rats.

As mentioned above, the HDAC inhibitor SAHA could attenuate sevoflurane-induced deficits in learning and memory in offspring. The present study attempted to investigate whether SAHA could also attenuate learning and memory deficits in offspring caused by maternal surgery under propofol anesthesia during mid-pregnancy.

## Materials and methods

The experimental protocol was approved by the Medical Research Ethics Committee of the Zhejiang Provincial People's Hospital Laboratory Animal Center(Protocol Number: A20220032). All animal experiments were performed according to the National Institutes of Health guide for the care and use of Laboratory animals (NIH Publications No. 8023, revised 1996). All surgery was performed under Propofol anesthesia, and all efforts were made to minimize suffering.

### Animals

Sprague-Dawley (SD) rats, 9–10 weeks old, weighing 265-305g, were purchased from zhejiang Provincial People's Hospital Laboratory Animal Center. SYXK(Zhe) 2019–0013, Hangzhou Zhejiang, China. After confirmation of pregnancy, the pregnant rats were identified and divided into propofol anesthesia group (Propofol group), surgery under propofol anesthesia group (Surgery group) and control group (Fig 1). All rats were housed separately under

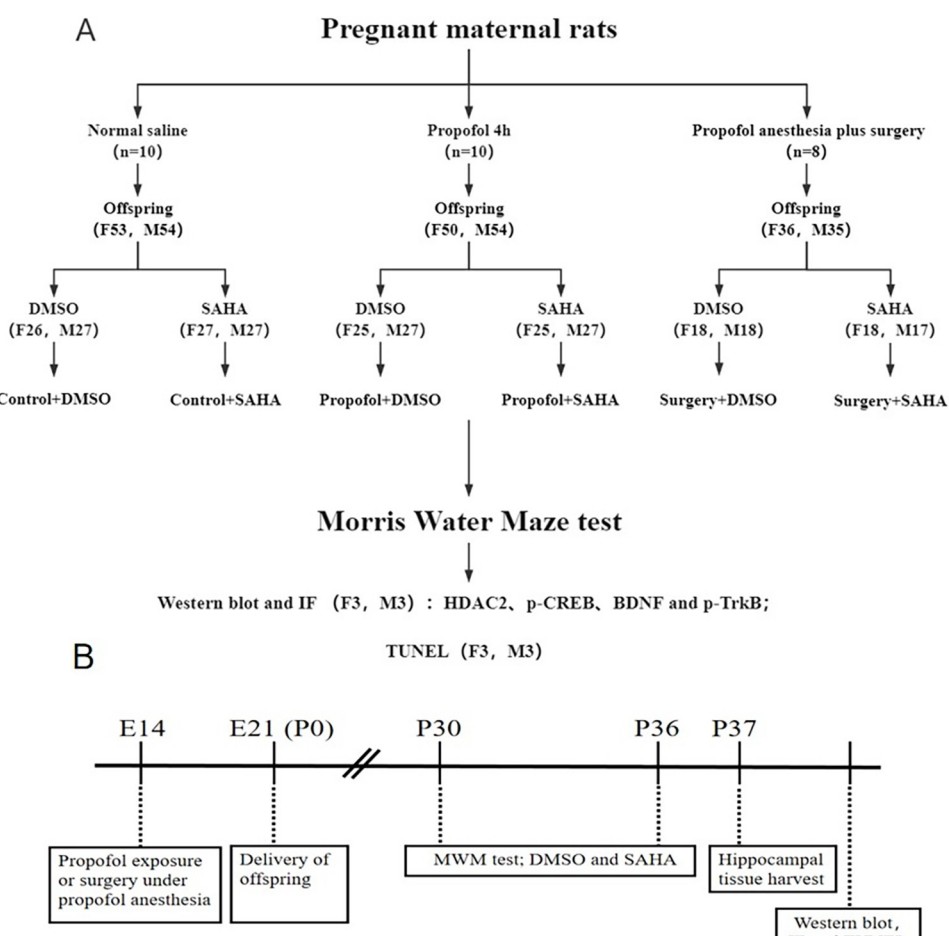

**Fig 1. The flow chart of experimental protocols. A)** The flow chart of the experimental protocols and distribution of offspring rats across different studies; **B)** The time-line of experimental paradigms. The number in brackets represents the number of animals. F, female; M, male; SAHA, HDAC2 inhibitor vorinostat; DMSO, dimethyl sulfoxide; IF, Immunofluorescence; TUNEL, terminal deoxynucleotidyl transferase mediated nick end labeling; E14, pregnant rats at gestational day 14; P0, postnatal day 0; ip, intraperitoneally.

standard laboratory conditions with a 12:12 light/dark cycle, 25±1°C, and 55±5% humidity, and they had free access to tap water and standard rat chow.

## Propofol exposure

Propofol exposure was conducted as in we previous report [28]. On day E14, a 24-gauge intravenous (IV) catheter was placed into the pregnant rat's lateral tail vein. Twenty mg/kg propofol (200 mg/20 ml, jc393, Diprivan, AstraZeneca UK Limited, Italy) was injected into the pregnant rats in the Propofol group or Surgery group via the IV catheter followed by 20 mg.kg-1.h-1 of continuous infusion for 4 hours after loss of right reflex. The dosage of anesthesia induction and the maintenance of propofol were selected based on our previous study [27, 28]. The pregnant rats in the control group received an equal volume of 20% intralipid instead of propofol.

## Surgery

Exploratory laparotomy was performed on the pregnant rats in the Surgery group. Anesthesia was induced and maintained with the same doses of propofol as used in the Propofol group.

The abdomen was shaved and sterilized with 70% sterile ethanol. An abdominal median incision (3 cm in length) was made after a subcutaneous injection of 0.125% bupicaine hydrochloride (0.2 ml per maternal rat). A normal saline-wetted sterile cotton swab was used to explore the abdominal cavity to see the diaphragmatic surface of the liver, the spleen, both kidneys, the bladder, etc. to mimic clinical exploratory laparotomy. The abdominal cavity was washed with 2 ml of 37˚C normal saline, followed by closure of the peritoneum, fasciae and abdominal musculature with 4–0 absorbable sutures. The skin was closed by 2–0 simple interrupted absorbable sutures. The procedure duration ranged from 20 to 30 minutes. The total time of propofol infusion was 4 hours. The maternal rats were returned to their cages after anesthesia recovery (return of the righting reflex) to continue their pregnancies.

## Monitoring

Electrocardiograms, pulse oxygen saturation (SpO2), heart rate, breath rate and noninvasive tail blood pressure were monitored during propofol infusion and surgery. Body temperature was monitored and maintained by a heating pad at 37˚C. If the cumulative duration of SpO2 falls below 95% and/or if there is a decrease in systolic blood pressure (SBP) exceeding 20% of baseline for more than 5 minutes, the maternal rat will be excluded from the study. A second rat will then be selected to ensure an adequate sample size, thereby eliminating any potential influence of maternal hypoxia or ischemia on the offspring.

## Arterial blood gases (ABG) analysis

To determine whether propofol exposure or surgery causes disturbances in mother's internal environment, another 18 pregnant rats were assigned to accept propofol, surgery under propofol anesthesia or act as a normal control (n = 6 per group). Femoral artery blood was collected at the end of the 4h propofol infusion or surgery to perform blood gases analysis and glucose detection.

## Drug administration

On postnatal day 30 (P30, which in rats corresponds to preschool age in humans (Rodier, 1980)), the offspring rats born to each mother rat from relative groups were randomly subdivided into dimethyl sulfoxide (DMSO) and SAHA. Two hours before each MWM trial, 90 mg/kg of SAHA (Selleck Chem, Houston, TX, USA) was intraperitoneally injected into the offspring in SAHA groups once per day for 7 consecutive days to investigate their effects on rat offspring's learning and memory. SAHA was dissolved in DMSO (Sigma Aldrich, Shanghai, China) solution, with final concentrations of 50 mg/ml. Equal volumes of DMSO solution were given to the offspring in the DMSO groups (Fig 1A).

## Morris water maze (MWM) task

The MWM system was used to evaluate the spatial learning and memory of offspring, as described in our previous studies [9, 26, 27]. A round steel pool, 150 centimeters (cm) in diameter and 60 cm in height was filled with water to a height of 1.0 cm above the top of a platform (15 cm in diameter). Water was kept at $(24 \pm 1)$˚C by an automatic thermostat (Beijing Sunny Instruments Co., Ltd., Beijing, China). MWM trial was performed once per day for 6 consecutive days starting day P30. Each rat was placed into the pool to search for the platform (located in the second quadrant, called the "target quadrant", with a clue on the inside wall of the pool) for 6 consecutive days, and the starting point (the third quadrant) was constant for each rat. When the rat found the platform, the rat was allowed to stay on it for 30 seconds (sec). If a rat

did not find the platform within 120 sec, the rat was gently guided to the platform and allowed to stay on it for 30 sec. The time for the rat to find the platform was named the "escape latency" (indicating learning ability). On the 7th day, the platform was removed, and the rat was placed in the same quadrant and allowed to swim for 120 sec. The number of times that the rats swam cross the area where the platform was previously hidden ("platform crossing times"), the time that the rat spent in the target quadrant ("target quadrant time"), the swimming trail and the speed of rats were recorded automatically and analyzed using MWM motion-detection software (Beijing Sunny Instruments Co., Ltd., Beijing, China) by a video tracking system. Both the platform crossing times and the target quadrant time reflected memory ability. The mean values of the escape latency, platform crossing times, target quadrant time and swimming speed of the offspring born to the same maternal rat were calculated as the final results. After each trial, the rat was cleaned with a dry towel and placed in a holding cage under a heat lamp until its hair dried before being returned to its cage.

## Hippocampal tissue harvest

Rats at day P37 were deeply anesthetized with an intraperitoneal injection of propofol and then killed by cervical dislocation. Hippocampal tissue was perfused transcardially with 0.9% saline and then soaked overnight in a cold 4% paraformaldehyde solution (in 0.1 M phosphate buffer, pH 7.4, 4° C). Hippocampal tissues were then embedded in paraffin for immunofluorescence (IF) and terminal deoxynucleotidyl transferase-mediated nick end labeling (TUNEL) staining. Hippocampal tissues for Western blotting were harvested only after trans myocardial perfusion with 0.9% cold saline and stored at -80°C.

## Western blot analysis

The hippocampi (6 offspring rats per group, male: female = 3:3) were homogenized on ice in RIPA lysis buffer (R0010, Beijing solar bio Co., Ltd., Beijing, China) containing a cocktail of protease inhibitors (DI111, Beijing TransGen Biotech Co., Ltd., Beijing, China) and a mixture of phosphatase inhibitors (P1260, APPLYGEN Gene Co., Ltd., Beijing, China). Protein concentration was determined by the bicinchoninic acid protein assay kit (P1511, APPLYGEN Gene Co., Ltd., Beijing, China). Protein samples (50 μg protein/lane) were separated by sodium dodecyl sulfate-polyacrylamide gel electrophoresis (SDS-PAGE) and transferred to a Polyvinylidene Fluoride (PVDF) membrane. The membranes were blocked by 5% nonfat dry milk tris buffered saline tween (TBST) for 1 hour and then incubated overnight at 4°C with relative primary antibodies: anti-HDAC2 antibody (1:1000, A19626, ABclonal, Wuhan, China), anti-p-CREB antibody (1:1000, AP0903, ABclonal, Wuhan, China), anti-BDNF antibody (1:500, ab108319, Abcam, Cambridge, MA, USA), anti-p-TrkB antibody (1:1000; Abcam, ab109684, Cambridge, MA, USA), and mouse anti-GAPDH (1:5000, Abcam, Cambridge, MA, USA). Thereafter, the membranes were washed three times with TBST buffer for 15 minutes, and the membranes were incubated with Goat Anti-Rabbit IgG (H+L), HRP Conjugate (1:1000, HS101, Beijing TransGen Biotech Co., Ltd., Beijing, China) or Goat Anti-Mouse IgG (H+L), HRP Conjugate (1:2000, HS201, Beijing TransGen Biotech Co., Ltd., Beijing, China) for 2 hours at room temperature. The membranes were washed three times with TBST buffer and detected using SuperSignal™ West Pico PLUS Chemiluminescent Substrate (34577, Thermo Fisher Scientific, Inc., Waltham, MA, USA). The images of the Western blot products were collected by a gel imaging system (BIO-RAD GelDoc 2000, Bio-Rad Laboratories, Inc. USA) and analyzed by Image-Pro Plus 6.0 (MEDIA CYBERNETICS, USA). The results were expressed per the integrated optical densities of the interesting protein relative to that of GAPDH. The results of offspring from all the other groups were then normalized to the average values of DMSO control offspring in the same Western blot.

## Immunofluorescence staining

The hippocampus sections (30 μm, 6 offspring rats per experimental group, 3 sections per animal) were incubated with 3% H2O2 for 25 minutes at room temperature in a wet box to inactive endogenous hydrogen peroxide enzymes. The sections were incubated with relative primary antibodies—anti-HDAC2 (1:200, ab32117, Abcam, Cambrige, UK), anti-Neun (1:200, ab177487, Abcam, Cambrige, UK), anti- p-CREB (1:100, ab32096, Abcam, Cambrige, UK)(dissolved in 1% goat serum albumin in phosphate buffered saline) at 4°C overnight. Then, the sections were exposed to the green fluorescent-conjugated secondary antibody and the red fluorescent-conjugated secondary antibody (1:500, TransGen Biotech, Beijing, China). Finally, the sections were wet mounted and immediately viewed using a fluorescence microscope (400X).

## Apoptosis assay

TUNEL staining was performed for paraffin sections using the In Situ Cell Death Detection Kit (Roche, Basel, Switzerland) according to the manufacturer's instructions. Briefly, after dewaxing and hydration, slices (6 offspring rats per experimental group, 3 slices per animal) were permeabilized in proteinase K (20 μg/ml) for 30 min at 37°C and then exposed to TUNEL reaction mixture for 2 hours at 37°C followed by incubation with a convertor-POD at 37°C for 30 min. Finally, the sections were incubated with diaminobenzidine substrate solution (DAB) for 15 min to visualize the TUNEL-positive cells and counterstained with hematoxylin for 30 sec. The TUNEL-positive cells (with deep brown stained nuclei) were observed under a light microscope at 400X magnification. The photos were taken, and the numbers of TUNEL-positive cells were counted with Image Pro Plus 6.0 (MEDIA CYBERNETICS, USA). Five visual fields were randomly selected for each section. The mean value of the TUNEL-positive cells ratio (the number of TUNEL positive cells/total cells x100%) was calculated as the final result.

## Statistical analysis

The nature of the hypothesis testing was two-tailed. All the results were assessed by well-trained investigators who were blind to group assignment. There were no missing data in this study. The sample size was based on our previous experience with this design [33]. The data are presented as mean±SD (standard deviations). The results of escape latency were subjected to two-way repeated measures analysis of variance (RM two-way ANOVA), followed by Bonferroni correction when a significant overall between-subject factor was found ($p < 0.05$). One-way analysis of variance (ANOVA) was used to analyze platform crossing times, target quadrant time, swimming speed, weight, average litter size, the expression levels of proteins (HDAC2, p-CREB, BDNF and p-TrkB) and apoptosis in the hippocampus followed by Bonferroni correction when a significant difference in groups was tested ($p < 0.05$). There were no outliers for any of the detected indexes. The survival rate and gender composition of the rat offspring were analyzed using the chi-square test. Statistical significance was considered when the value of $p < 0.05$. The statistical analysis software was SPSS version 17.0 (IBM, UK).

## Results

### Arterial blood gases (ABG) and glucose of the pregnant rats

At the end of propofol or surgery exposure, ABG and glucose were detected. The results showed no differences in blood gas and blood glucose levels in the pregnant rats across the Control, Propofol and Surgery groups (Table 1).

**Table 1. Comparisons of maternal arterial blood gas and glucose levels.**

| ABG | Control group | Prop group | Surg group |
|---|---|---|---|
| pH | 7.35 ± 0.06 | 7.33 ± 0.09 | 7.37 ± 0.07 |
| $PaO_2$ (mmHg) | 98.66 ± 1.50 | 98.16 ± 2.63 | 96.67 ± 2.59 |
| $PaCO_2$ (mmHg) | 42.00 ± 2.09 | 42.33 ± 1.96 | 42.00 ± 2.52 |
| $HCO_3^-$ (mmol/L) | 25.33 ± 2.31 | 25.28 ± 3.21 | 26.50 ± 2.94 |
| BE (mmol/L) | 2.56 ± 0.26 | 2.44 ± 0.27 | 2.45 ± 0.49 |
| $Na^+$ (mmol/L) | 139.83 ± 1.83 | 140.66 ± 1.96 | 140.00 ± 1.41 |
| $K^+$ (mmol/L) | 3.68 ± 0.11 | 3.70 ± 0.23 | 3.70 ± 0.10 |
| $Ca^{2+}$ (mmol/L) | 1.33 ± 0.04 | 1.34 ± 0.10 | 1.31 ± 0.02 |
| Glucose (mmol/L) | 9.46 ± 0.88 | 9.75 ± 0.10 | 9.78 ± 0.90 |

Data are expressed as means ± SD. n = 6 for each group.

### Physical characteristics of the offspring rats

The body weight of the rat offspring was evaluated on P30. There was no difference in the average body weight, average litter size, survival rate (the ratio of rat offspring that survived past day P30), or Male and female distribution of offspring among the Control, Propofol, and Surgery groups (Fig 2A–2D). No dyskinesia was observed in the rat offspring (evaluated by daily inspection and the swimming speed of the rat offspring in the MWM tests).

### Deteriorating effect of surgery on offspring's learning and memory

Learning and memory abilities in the offspring rats were evaluated using the MWM system from P30 through P36. The results showed that propofol exposure increased the time to find

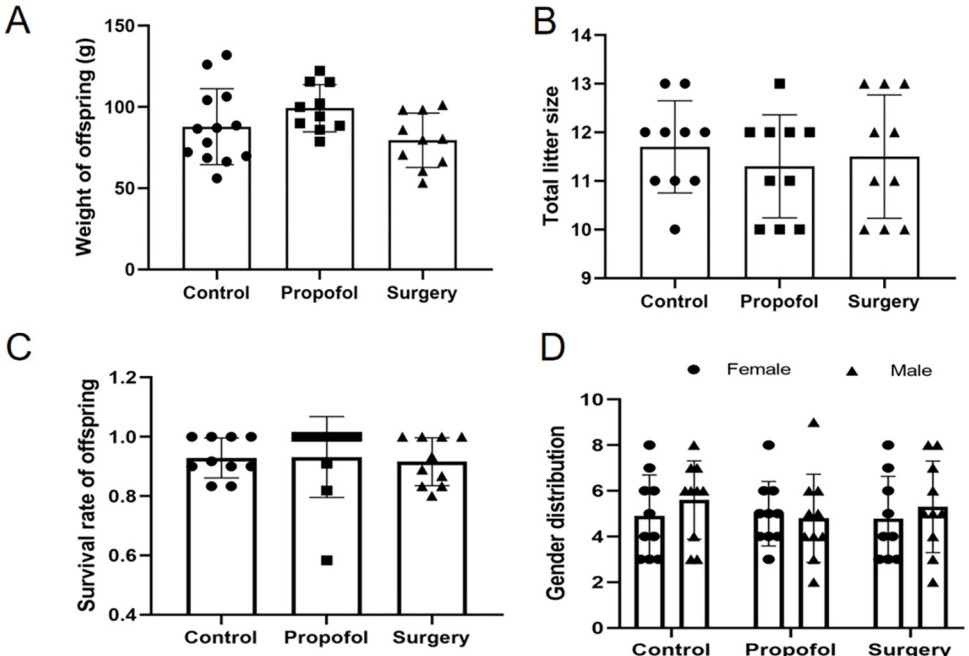

**Fig 2. The physical characteristics of rats' offspring. A**) Body weight of offspring rats; **B**) Total litter size in each group; **C**) Survival rate of offspring rats (defined as the ratio of rat offspring that survived over P30 day); **D**) Gender composition in each group. There was no significant difference in these indexes among the control, propofol and surgery groups. The data are expressed as means ± SD.

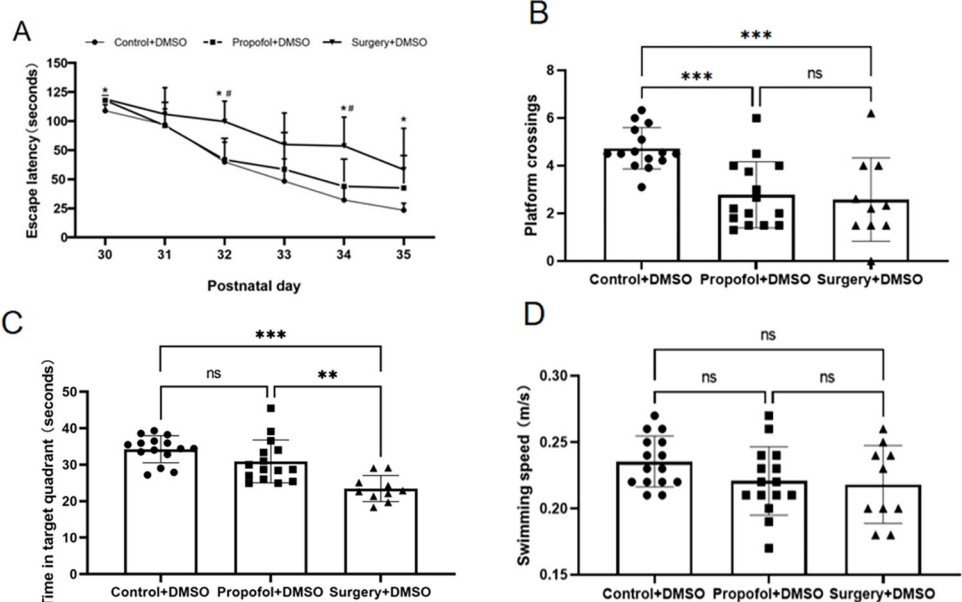

**Fig 3. Surgery exacerbates maternal propofol exposure-induced deficits in water maze learning and memory. A**) Escape latency (indicating learning ability): the offspring rats in the Propofol+DMSO group had a comparable escape latency with those in the Control+DMSO group. However, the offspring rats in the Surgery+DMSO group showed a significantly longer escape latency (*$p < 0.05$ vs. Control+DMSO), and with those offspring rats in the Propofol +DMSO group (#$p < 0.05$ vs. Surgery +DMSO). **B**) The platform crossing times (indicating memory ability): the offspring rats in both the Propofol+DMSO group and the Surgery+DMSO group had significantly less platform crossing times (*$p = 0.001$ vs. Control+DMSO). **C**) Target quadrant time (indicating memory ability): There was no statistical difference in target quadrant time between the offspring rats in the Propofol+DMSO and Control+DMSO groups. However, the offspring rats in the Surgery+DMSO group spent significantly less time in the target quadrant (*$p<0.001$ vs. Control+DMSO; *$p<0.01$ vs. Propofol+DMSO group). **D**) Swimming speed: there was no statistical difference in swimming speed among the three groups. The data are presented as means ± SD. Control+DMSO group, $n = 15$; Propofol+DMSO group, $n = 15$; Surgery+DMSO group, $n = 10$.

the platform (escape latency). When combined with surgery, the escape latency was increased significantly, especially on P32 and P34 (Fig 3A). Meanwhile, both propofol exposure and surgery decreased the platform crossing times and target quadrant time (an index for memory ability), and the surgery decreased more significantly (Fig 3B and 3C). There was no significant difference in the offspring's swimming speed across groups (Fig 3D). After being treated with SAHA, the escape latency in propofol and propofol plus surgery exposed rat offspring's were shortened, meanwhile, both of the platform crossing times and target quadrant time were increased (Figs 4 and 5). But SAHA did not affect their swimming speed (Figs 4D and 5D). SAHA did not affect these indexes in rat offspring that had not exposed to propofol or surgery (Fig 6).

## Over-expression of HDAC2 protein caused by propofol and surgery

To determine whether HDAC2 is involved in the learning and memory impairment caused by maternal propofol exposure or surgery, the expression of HDAC2 protein in rat offspring's hippocampus was detected by immunofluorescence (IF) staining and Western blotting. We observed the co-localization of the typical neuronal biomarker Neun, a marker that can be accurately and stably expressed in the nucleus, with HDAC2 by IF staining. The current findings demonstrated that the nucleus of hippocampus neurons was the primary site of HDAC2 expression. The present study showed that HDAC2 has no regional specificity. (Fig 7A). The

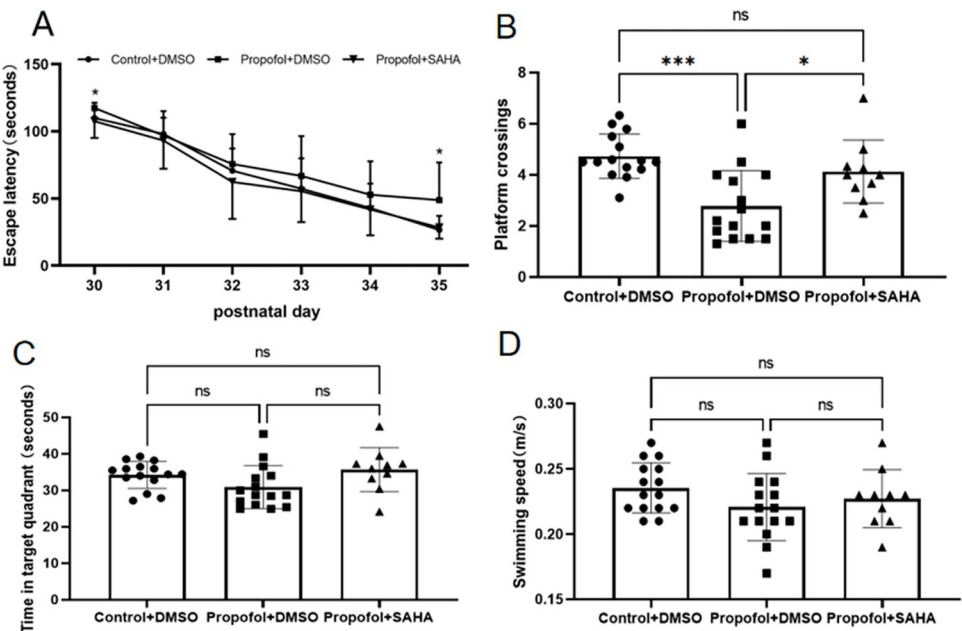

**Fig 4. SAHA rescues the learning and memory deficits caused by propofol. A)** The offspring rats in the Propofol +DMSO group had significantly longer escape latency than those offsprings in the Control+DMSO group. However, such an effect was rescued by SAHA (see the Propofol+SAHA group), especially on days P30 and P35 (*$p < 0.05$ vs. the Propofol+DMSO group). **B)** The offspring rats in the Propofol+DMSO group showed significantly less platform crossing times (see the Propofol+DMSO group; *$p < 0.001$ vs. Control+DMSO), and such effect was rescued by SAHA (see the Propofol+SAHA group; *$p < 0.05$ vs. Propofol+DMSO). **C)** There was no statistical difference in target quadrant time among the three groups of offspring rats. **D)** There was no statistical difference in swimming speed among the three groups of offspring rats. The data are presented as mean ± SD. Control+DMSO group, n = 15; Propofol+DMSO group, n = 15; Propofol+SAHA group, n = 10.

results of Western blotting showed that maternal propofol exposure increased the level of HDAC2 protein in rat offspring's hippocampus (Fig 7B and 7C), whereas surgery under propofol anesthesia induced a much more significant increase of HDAC2 protein (Fig 7B and 7C). SAHA treatment ameliorated the overexpression of HDAC2 induced by propofol or surgery exposure (Fig 7B and 7C), but had no effect on the expression of HDAC2 in the rat offspring that had not exposed to propofol or surgery (Fig 7B and 7C) (raw data see S2 Fig 7 in S1 Raw images).

## Downregulated expression of p-CREB caused by propofol and surgery

Immunofluorescence staining revealed that p-CREB was mainly expressed in the nuclei of hippocampal neurons. Both the number of p-CREB positive cells and the fluorescence intensity were decreased after propofol anesthesia or surgery exposure (Fig 8A). Western blotting showed that propofol anesthesia alone downregulated the expression of p-CREB protein, and surgery under propofol anesthesia further reduced the expression of p-CREB protein. SAHA mitigated the downregulation of p-CREB expression induced by propofol anesthesia or surgery exposure significantly (Fig 8B and 8C) (raw data see S3 Fig 8 in S1 Raw images).

## Disturbance of BDNF-TrkB signaling pathway

Maternal propofol exposure downregulated the expression of the BDNF and p-TrkB proteins in the rat offspring's hippocampi, and surgery under propofol anesthesia further

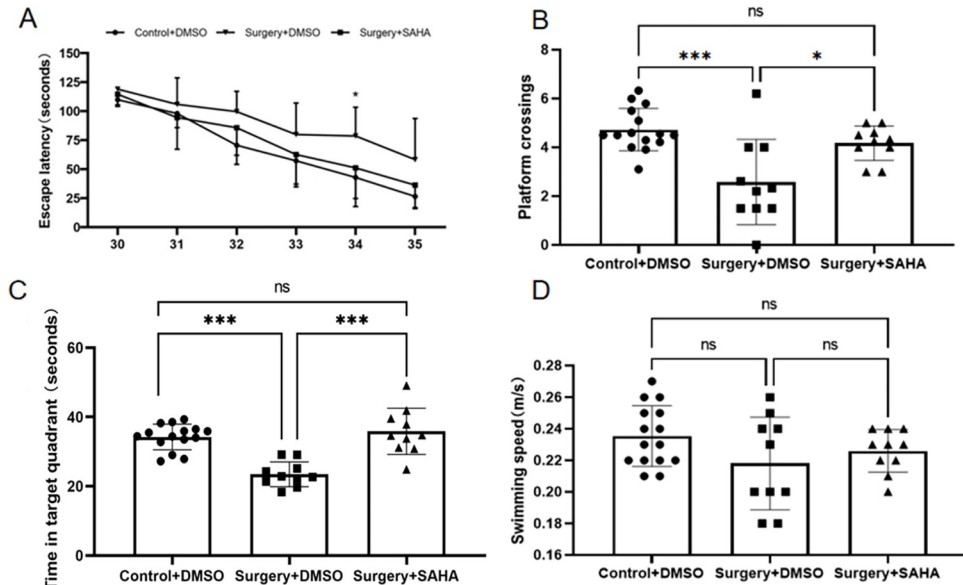

**Fig 5. SAHA rescues the learning and memory deficits caused by surgery. A**) The offspring rats in the Surgery +DMSO group had relatively longer escape latency than those in the Control+DMSO group. However, the offspring rats in the Surgery+SAHA group had significantly less escape latency, especially on day P34 (*$p < 0.05$ vs. Surgery +DMSO). **B**) The offspring rats in the Surgery+DMSO group showed relatively less platform crossing times than those in the Control+DMSO group(see the Surgery+DMSO group; *$p<0.001$ vs. Control+DMSO). However, the offspring rats in the Surgery+SAHA group had significantly more platform crossings (see the Surgery+SAHA group; *$p < 0.05$ vs. Surgery+DMSO). **C**) The offspring rats in the Surgery+DMSO group spent relatively less time in the target quadrant than those in the Control+DMSO(see the Surgery+DMSO group;*$p<0.001$ vs. Control+DMSO). However, the offspring rats in the Surgery+SAHA group spent a significantly longer time in the target quadrant (see the Surgery +SAHA group;*$p<0.001$ vs. Surgery+DMSO). **D**) There was no statistical difference in swimming speed among the three groups of offspring rats. The data are presented as means ± SD. Control+DMSO group, n = 15; Surgery+DMSO group, n = 15; Surgery+SAHA group, n = 10.

downregulated their expression. Upon treatment with SAHA, the levels of both BDNF and p-TrkB protein were restored significantly (Fig 9) (raw data see S4 Fig 9 in S1 Raw images).

## Apoptosis of hippocampal neurons after surgery

Both propofol anesthesia and surgery under propofol anesthesia-induced hippocampal neuronal apoptosis in offspring, but surgery under propofol anesthesia resulted in more severe neuronal apoptosis (Fig 10). SAHA treatment had no effect on the apoptosis induced by propofol anesthesia or surgery under propofol anesthesia (Fig 10).

## Discussion

The present study showed that maternal propofol exposure during middle pregnancy causes learning and memory deficit, overexpression of hippocampal HDAC2 and neuronal apoptosis, and downregulation of hippocampal p-CREB, BDNF and p-TrkB in offspring rats. Surgery causes more significant changes to these indexes. SAHA reverses the learning and memory impairments and the changes of HDAC2, p-CREB, BDNF and p-TrkB protein expression levels induced by propofol or surgery under propofol anesthesia, but could not ameliorate the hippocampal neuronal apoptosis induced by propofol or surgery. These results suggest that SAHA may alleviate learning and memory impairments caused by maternal propofol anesthesia or surgical exposure through certain signaling pathways.

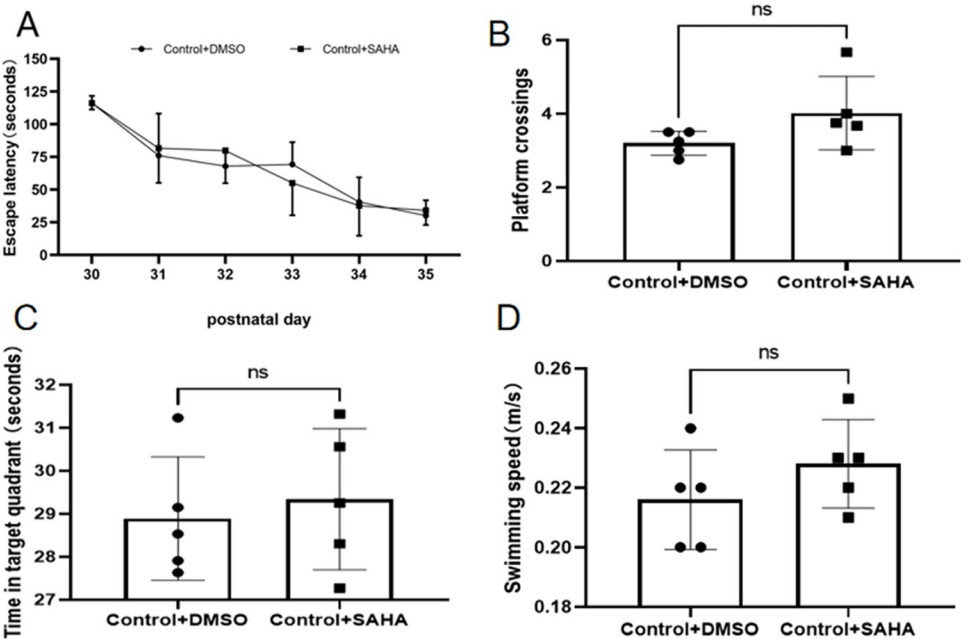

**Fig 6. SAHA produced no effect on the learning and memory of normal control offspring rats. A)** There was no difference in escape latency between the offspring rats in the Control+SAHA and Control+DMSO groups. **B)** There was no difference in platform crossings between the two groups. **C)** There was no difference in time spent in target quadrant between the two groups. **D)** There was no difference in swimming speed between the two groups. The data are presented as means ± SD. Control group, n = 5. Control+SAHA, n = 5.

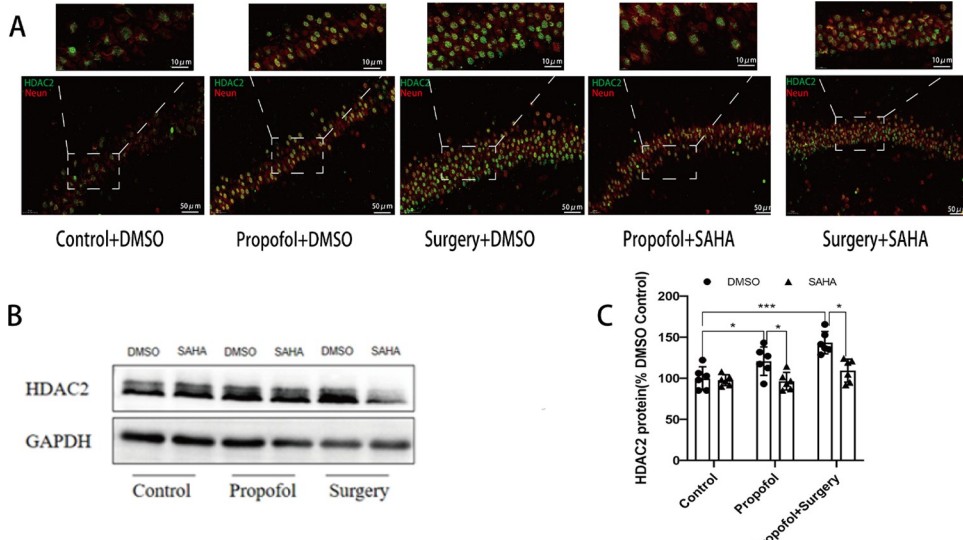

**Fig 7. Propofol anesthesia or with surgery enhanced the expression of HDAC2 and SAHA reversed the enhancement. A)** Immunofluorescence images for the distribution of HDAC2-positive cells and neun neurobiomarkers in the hippocampus. (green represents HDAC2 and red represents Neun). **B)** Western blotting images for HDAC2 protein expression in the hippocampus. **C)** There was a significant increase in the protein level of HDAC2, especially in the propofol anesthesia plus surgery. (*$p < 0.001$ vs. Control). SAHA reversed the elevation of HDAC2 protein levels induced by propofol anesthesia or propofol anesthesia plus surgery. (*$p < 0.05$ vs. DMSO). The data are presented as means ± SD. n = 6 per group; female:male = 3:3.

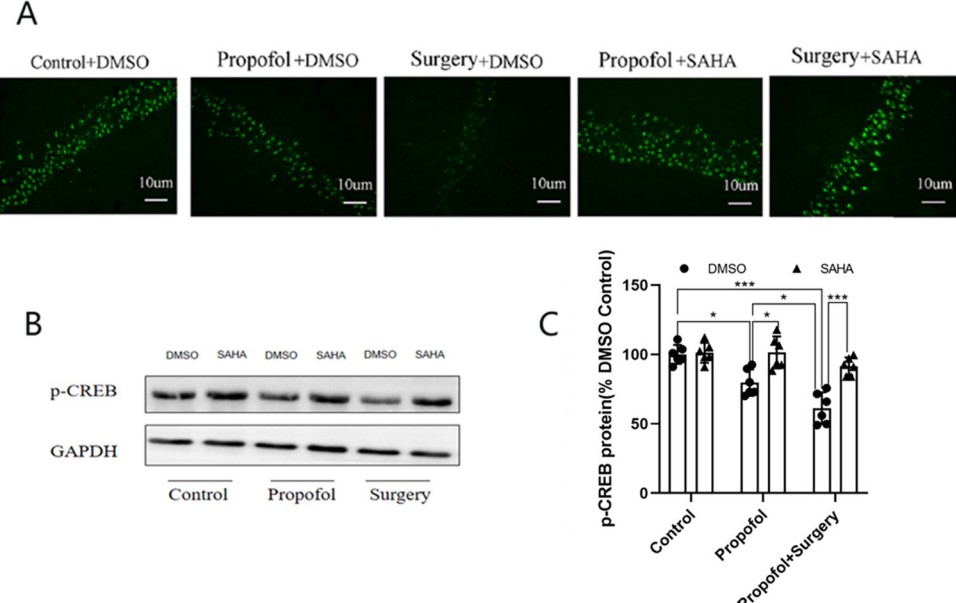

**Fig 8. Propofol anesthesia or with surgery decreased the expression of p-CREB and SAHA reversed the reduction. A)** Immunofluorescence image for the distributive expression of p-CREB. **B)** Western blotting images for p-CREB protein. **C)** There was no difference between the Control+SAHA and the Control+DMSO. The protein levels of p-CREB were downregulated expression in the Propofol+DMSO (*$p < 0.05$ vs. Control+DMSO group) and Surgery +DMSO (*$p < 0.001$ vs. Control+DMSO group). Surgery under propofol anesthesia decreased the expression more significantly. (*$p < 0.05$ vs. Propofol+DMSO group). SAHA reversed the decreased expression of p-CREB protein levels induced by propofol anesthesia or propofol anesthesia plus surgery (*$p < 0.001$ vs. DMSO group). The data are presented as means ± SD. n = 6 per group, female: male = 3:3.

No difference in vital signs, artery blood gases, or blood glucose levels was observed across the groups. Therefore, the impaired learning and memory may not be caused by physical differences but caused by propofol anesthesia or surgery itself. Previous studies suggested a sex-specific sensitivity to general anesthesia [29]. In the present study, there was no significant difference in sex composition among all groups, suggesting that the learning and memory deficits observed in the present study were not caused by differences in sex.

Long-term potentiation (LTP) plays an important role in memory formation [30] HDAC2 is one of the members of histone deacetylases, which plays a critical role in histone acetylation/deacetylation processes. Loss of HDAC2 gene improves working memory [31]. SAHA could normalize the impaired contextual fear conditioning in HDAC2 overexpressed mice but has no effect in HDAC2-deficient mice [20], indicating that SAHA needs to work based on HDAC2 background. The previous study in our laboratory showed that intraperitoneal injection of SAHA (90 mg/kg; 2 hours before each daily session of MWM training for 7 consecutive daily sessions) could ameliorate offspring rats' learning and memory deficit induced by maternal isoflurane exposure during late-stage of pregnancy [9] or by propofol exposure during early gestation [26]. The present study showed that maternal propofol exposure during middle pregnancy induced overexpression of HDAC2 in offspring rat's hippocampi, and such overexpression was further enhanced upon surgical operation. After treatment with SAHA, the overexpression of HDAC2 was reduced and the impaired learning and memory were rescued. Thus, the learning and memory impairment caused by maternal propofol anesthesia or surgery was associated with the overexpression of HDAC2.

HDAC2 contributes to synaptic plasticity by regulating the transcriptional activation of CREB. It has been documented that CREB deficiency impairs LTP and spatial memory

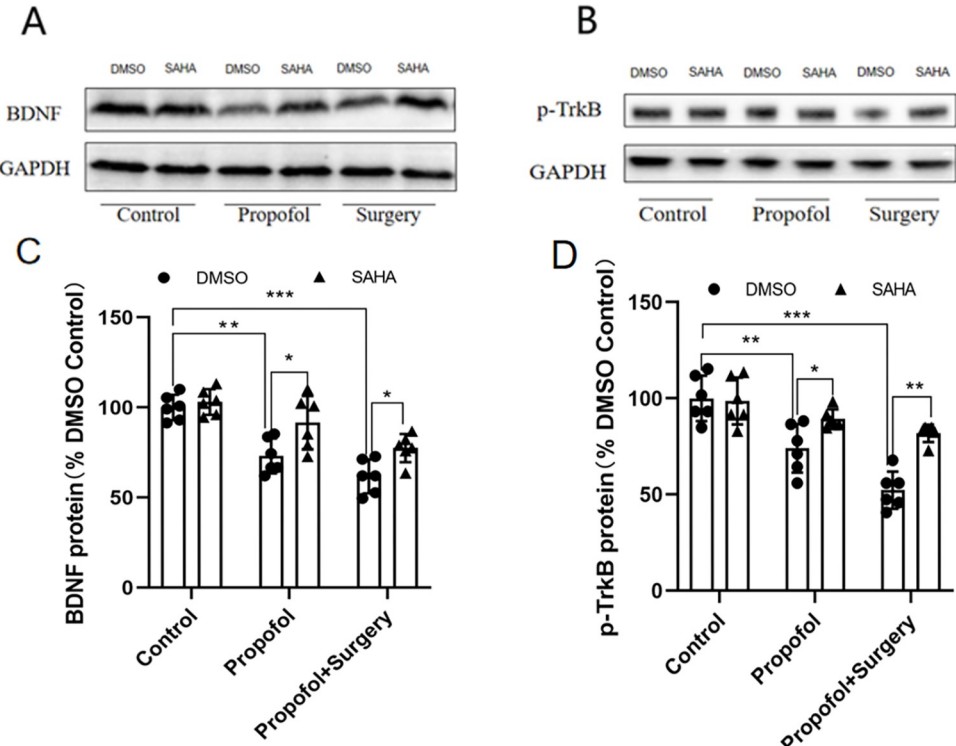

**Fig 9. Propofol anesthesia or with surgery decreased the expression of BDNF and p-TrkB, but SAHA reversed the reduction. A**) Western blotting bands of BDNF. **B**) Western blotting bands of p-TrkB. **C**) BDNF protein levels (BDNF protein %DMSO Control): propofol exposure and surgery significantly decreased the expression level of BDNF. Compared with Control+DMSO group, (*$p < 0.01$ vs. Propofol+DMSO, *$p < 0.001$ vs. Surgery+DMSO group). SAHA alleviated the decrease caused by propofol or propofol anesthesia plus surgery significantly. Compared with the corresponding DMSO group, (*$p < 0.05$ vs. Propofol+SAHA; *$p = 0.05$ vs. Surgery+SAHA). **D**) p-TrkB protein levels in rat offspring's hippocampus: propofol anesthesia decreased p-TrkB protein levels significantly. While propofol anesthesia plus surgery, the levels of p-TrkB protein decreased much more significantly. Compared with Control +DMSO group, (*$p < 0.01$ vs. Propofol+DMSO group; *$p < 0.001$ vs.Surgery+DMSO group). SAHA alleviated the decrease of p-TrkB protein levels caused by propofol anesthesia or surgery significantly, though the levels of p-TrkB protein in Surgery+SAHA were still lower than Control+DMSO group. Compared with the corresponding DMSO group, (*$p < 0.05$ vs. Propofol+SAHA, *$p < 0.01$ vs. Surgery+SAHA). **Note:** the data are presented as the mean ± SD. n = 6 in each group, female: male = 3:3.

consolidation [12]. On the other hand, enhanced phosphorylation of CREB alleviates learning and memory impairment [32], and decreased phosphorylation of CREB impairs long-term spatial memory [33]. It has been reported that rescue of the CREB-protein-signaling pathway reverses the impairments of spatial memory retention caused by subclinical doses of propofol in adult rats [34]. Previous study in our laboratory showed that maternal exposure to isoflurane or propofol during pregnancy impaired learning and memory in offspring rats by downregulating the expression of CREB [9]. The present study showed that propofol exposure reduced hippocampal p-CREB levels of offspring rats, and the reduction of p-CREB levels was exacerbated upon surgery. These changes in p-CREB level were rescued upon SAHA treatment.

Synaptophysin provides a structural basis for synaptic plasticity [35] and modifies synaptic plasticity through the BDNF-TrkB signaling pathway [36]. A previous study in our laboratory showed that propofol exposure during late pregnancy caused persistent deficits in learning and memory in offspring rats via the BDNF-TrkB signaling pathway [27]. The present study showed that maternal propofol exposure during middle pregnancy reduced the levels of BDNF

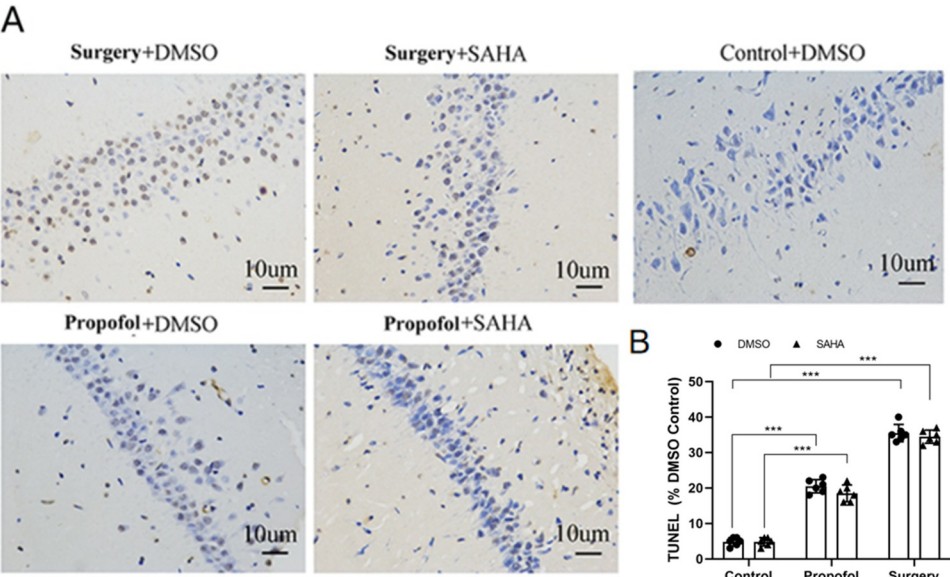

**Fig 10. Surgery resulted in more severe neuronal apoptosis, but SAHA had no effect on the apoptosis. A**) TUNEL staining for neuronal apoptosis in the hippocampus of rat offspring. **B**) Apoptosis ratio (TUNEL positive cells / total neurons ×100%), Propofol significantly induced neuronal apoptosis in rat offspring's hippocampus. Surgery induced much more neuronal apoptosis than Propofol anesthesia alone. Compared with Control+DMSO group (*$p < 0.001$ vs. Propofol+DMSO group, *$p < 0.001$ vs. Surgery+DMSO group). Compared with Control+SAHA group, (*$p < 0.001$ vs. Propofol+SAHA group, *$p < 0.001$ vs. Surgery+SAHA group). **Note:** the data are presented as the mean ± SD. n = 6 for each group, female: male = 3:3.

and p-TrkB and such reduction was exacerbated upon surgery. Treatment with SAHA rescued the learning and memory deficit and the downregulated expression of BDNF and p-TrkB in the hippocampus. Our results confirmed that the BDNF-TrkB signaling pathway is involved in the learning and memory impairments caused by maternal propofol exposure or surgery under propofol anesthesia.

Histone acetylation is tightly co-regulated by the opposing effects of histone acetyltransferase (HAT) and HDAC [12, 13]. Therefore, the overall effects of inhibiting HAT and activating HDAC could deacetylate lysine and then inhibit the transcription of genes [37]. Consistently, the present study found that the expression levels of hippocampal, BDNF, and p-TrkB were reduced in the offspring rats receiving either propofol anesthesia or surgery under propofol anesthesia. It remains to be confirmed if such effects are directly due to decreased expression of HAT and increased expression of HDAC.

Growing evidence demonstrates that propofol exposure increases neuroapoptosis in the hippocampus and results in cognitive dysfunctions [4, 24, 38] depending on the dose, time, and timing of the exposure, and on the anesthetics and drug combinations as well [39]. The present study demonstrated that exposure to propofol during mid-pregnancy induces neuronal apoptosis in the hippocampi of offspring, and surgical intervention exacerbates this effect. These findings are consistent with previous studies indicating that intraperitoneal injection of propofol or surgery under propofol anesthesia on postnatal day 7 in offsprings leads to neuronal apoptosis and subsequent long-term cognitive dysfunction in adulthood [40], and surgery modifies the effects of general anesthetics on neuronal structure [34]. It is reported that SAHA could inhibit seizure-induced hippocampal neuronal apoptosis in developing rats. However, the present study showed that SAHA could not rescue the effect of propofol exposure or propofol exposure plus surgery on hippocampal neuronal apoptosis.

Increasing evidence suggests that short-term exposure to a low dose of anesthetic produces a neuroprotective effect on the developing brain, whereas prolonged exposure to a high dose of anesthetic results in cognitive dysfunction [41, 42]. The previous study conducted in our laboratory also demonstrated that exposure to propofol during early gestation, at the same dosage as used in the present study, did not elicit any discernible effects on hippocampal learning and memory in offspring rats when the exposure duration was limited to 2 hours. However, a prolonged exposure time of 4 or 8 hours induced significant deficits in learning and memory. Whether the impact of propofol anesthesia during mid-pregnancy on hippocampus-dependent learning and memory in offspring rats is contingent upon dosage or duration of exposure needs further study.

The combination of propofol anesthesia and surgery is a routine clinical condition. It is widely acknowledged that perioperative neurocognitive disorders (for example, learning and memory deficits) are associated with specific types of surgical procedures [43]. This kind of impairment could be due to inflammation. It has been reported that surgery activates a systematic cascade of inflammatory signaling molecules [44] and triggers local inflammatory responses. The postoperative elevation of inflammatory factors such as TNF-α, IL-6, IL-1β, and HMGB-1 could impair hippocampus-dependent learning and memory [45], resulting in postoperative cognitive dysfunction (POCD) [46]. Further investigations are needed to explore the molecular and cellular mechanisms underlying the impact of maternal surgical procedures on offspring's cognitive function.

There were limitations in the present study. We did not examine hippocampal synaptic plasticity using a neurophysiological approach and did not detect the pathological changes of neurons in the fetal brains immediately after maternal propofol anesthesia or surgery. The causal relationship between the expression changes in the observed proteins and the deficits in learning and memory behavior remain to be confirmed. Furthermore, the possible effects of maternal propofol exposure or surgery under propofol anesthesia on other brain regions (such as the cerebral cortex) of the offspring were not examined.

## Summary and conclusion

The present study demonstrates that maternal nonobstetric surgery during mid-pregnancy exacerbates hippocampus-dependent spatial learning and memory impairment in offspring rats caused by propofol anesthesia, which is associated with increased expression of HDAC2 and decreased levels of synapse-associated proteins p-CREB, and BDNF-TrkB. Treatment with SAHA could rescue the learning and memory deficits and the alterations in synapse-associated proteins induced by maternal surgery under propofol anesthesia in offspring. Thus, SAHA could be a potential and promising agent in clinical application.

## Supporting information

**S1 Raw images.**
(PDF)

## Acknowledgments

The authors express sincere thanks to Professors Guangqin Fan and Chang Feng, both affiliated with the Jiangxi Provincial Key Laboratory of Preventive Medicine at Nanchang University (Nanchang, China), for their helpful discussion and technical assistance. We owe our great thanks to Professor Baoming Li at the Hangzhou Normal University Institute of Brain Science for his guidance in data analysis and manuscript organization. We also thanks to Professor

Yiping Mou, the general Surgery, Cancer Center, Department of Gastrointestinal and Pancreatic Surgery, Zhejiang Provincial People's Hospital (Affiliated People's Hospital), Hangzhou Medical College, for his providing experimental technical support.

## Author Contributions

**Conceptualization:** Yunlin Feng, Yanfei Lu, Foquan Luo.

**Data curation:** Yunlin Feng, Jia Qin, Mengdie Wang.

**Formal analysis:** Jia Qin, Shengqiang Wang.

**Funding acquisition:** Foquan Luo.

**Methodology:** Yanfei Lu, Shengqiang Wang.

**Writing – original draft:** Yunlin Feng, Jia Qin.

**Writing – review & editing:** Foquan Luo.

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
