## [Decision Letter · Decision Letter 0]

18 Dec 2023

PONE-D-23-37577Suberoylanilide Hydroxamic Acid Attenuates Cognitive Impairment in Offspring Caused by Maternal Surgery during Mid-pregnancyPLOS ONE

Dear Dr. Luo,

Thank you for submitting your manuscript to PLOS ONE. After careful consideration, we feel that it has merit but does not fully meet PLOS ONE’s publication criteria as it currently stands. Therefore, we invite you to submit a revised version of the manuscript that addresses the points raised during the review process.

The reviewers provide a variety of helpful comments regarding the presentation and discussion of the data.  When preparing your revised version, please take them into account.  Furthermore, for all bar graph presentations I recommend that you also include the individual data points, a mode of presentation that is becoming the standard in the field.  Furthermore, there is some language editing required.  You may consider using an English language editing service.

We look forward to receiving your revised manuscript.

Kind regards,

Uwe Rudolph

Academic Editor

PLOS ONE

Journal Requirements:

 the National Natural Science Foundation of China (81960211, 81460175 and 81060093) and Zhejiang medicine and health science and technology project of Zhejiang Provincial Health Commission (2023KY037).

Reviewers' comments:

Reviewer's Responses to Questions

**Comments to the Author**

1. Is the manuscript technically sound, and do the data support the conclusions?

Reviewer #1: Yes

Reviewer #2: Yes

2. Has the statistical analysis been performed appropriately and rigorously? 

Reviewer #1: Yes

Reviewer #2: Yes

3. Have the authors made all data underlying the findings in their manuscript fully available?

Reviewer #1: Yes

Reviewer #2: No

4. Is the manuscript presented in an intelligible fashion and written in standard English?

Reviewer #1: Yes

Reviewer #2: Yes

5. Review Comments to the Author

Reviewer #1: The manuscript investigate the effect of SAHA on propofol-induced learning and memory deficits in offspring caused by maternal surgery during mid-pregnancy. They clarified maternal propofol exposure during middle pregnancy impaired the water-maze learning and memory of the offspring rats, increased the protein level of HDAC2 and reduced the protein levels of p-CREB,BDNF and p-TrkB in the hippocampus of the offspring, and such effects were exacerbated by surgery. SAHA alleviated the cognitive dysfunction and rescued the changes in the protein levels of p-CREB, BDNF and p-TrkB induced by maternal propofol exposure alone or maternal propofol exposure plus surgery. In general, the flow of the experiments is logical, and the findings are of considerable interest to the field. Based on the current results, the authors concluded that SAHA could be a potential and promising agent for treating the learning and memory deficits in offspring caused by maternal nonobstetric surgery under propofol aneshtesia. However, the current dataset should be enriched with a few supplemental details.

1.In the fig4, the platform crossing times in the Propofol +DMSO group was less than that in the Control+DMSO group? There was no label for the statistical analysis of the two group. Please further verify the original data and related statistical methods.

2.In the fig7, IF staining results revealed that HDAC2 predominantly expressed in the hippocampal neuronal nucleus. Did the expression of HDAC2 in the hippocampus appear regional specificity? The HDAC2 should be detected by staining with neuronal biomarker.

Reviewer #2: This is a straightforward manuscript from Feng et al., entitled “Suberoylanilide Hydroxamic Acid Attenuates Cognitive Impairment in Offspring Caused by Maternal Surgery during Mid-pregnancy.” The authors based their work on the premise that maternal exposure to commonly used anesthesia (during obstetric and non-obstetric interventions) causes cognitive deficits in offspring, and that decreased histone acetylation has been shown to induce deficits in hippocampus dependent learning and memory. The paper demonstrates that treatment with the class 1 and 2b HDAC inhibitor SAHA can rescue the cognitive deficits and the alterations in synapse-associated proteins induced by maternal surgery under propofol anesthesia in offspring. The data presented are convincing, however a key piece of data that appears missing is the effect of propofol on some histone or non-histone acetylation marks linked to memory. Does propofol decrease acetylation? If so, SAHA would likely reverse such effects.

The following comments and suggestions should improve the quality of the manuscript.

• The bar graphs in figure 2 should have scattered points for the reader to better understand the spread of the data.

• Scattered points should also be added to Figures 3B, 3C and 3D.

• Scattered points should be added to figures 4B, 4C and 4D.

• It is not clear what significance is indicated in figure 4B for "propofol + SAHA". Is it a comparison between "propofol + DMSO" and "propofol + SAHA"? If so, it should be clearly indicated on the graph with brackets or other means.

• It would also be good to indicate whether the difference between "control + DMSO" and "propofol + DMSO" is significant. By eye it looks as significant if not more than "propofol + DMSO" vs "propofol +SAHA."

• The same corrections as in figure 4 are needed for figure 5. More detailed graphical representations of the significance are needed for figures 5B and 5C.

• Scattered points are needed for Figures 6B, 6C and 6D. Whenever there appears to be a slight difference, the stats should be graphically shown. For example, it could be "*" for a significance of p<0.05 or "ns" whenever p>0.05.

• Are figure 7C data normalized to DMSO control? If so the legend in the Y-axis should say % DMSO control. The same comment goes for figures 8C, 9C, 9D and 10B.

• Figure 7C graphically presents the stats well. It needs to be presented, however as bar graphs with scattered points. The same goes for figures 8C, 9C, 9D and 10B.

• The western blots used for quantification for figure 7C should be provided in the supplementary material. The same goes for Figures 8C, 9C and 9D.

• Do the authors have a hypothesis about why surgery exacerbates the effects of propofol? This should be addressed in the discussion. Could it be inflammation related?

• Editing of the manuscript is needed. for example, the word "purchased" is erroneously separated as "pur" on line 105 and "chase" on line 106. Similarly, there is "laborator" on line 110 and "y" on line 111 to spell out "laboratory". There are many other instances of the same type of error. There are also many places where there is no space after periods, when a new sentence is starting.

6. PLOS authors have the option to publish the peer review history of their article (what does this mean?). If published, this will include your full peer review and any attached files.

Reviewer #1: No

Reviewer #2: No

---

## [Author Response · Author response to Decision Letter 0]

28 Jan 2024

Dear Editor,

We appreciate for the editor's valuable suggestions. We have incorporated individual data points into all bar graph presentations, as suggested. Furthermore, we have addressed some language issues in our revised manuscript.

Dear Reviewer1,

Thank you very much for the positive and constructive comments/suggestions on our manuscript. We have carefully revised the manuscript according to your comments and suggestions, and hope our revisions could meet your requirements. 

Dear Reviewer2,

We greatly appreciate your insightful and helpful feedback on our paper. We hope that our careful changes, which took into account your feedback and recommendations, would meet your requirements.

---

## [Decision Letter · Decision Letter 1]

15 Feb 2024

Suberoylanilide Hydroxamic Acid Attenuates Cognitive Impairment in Offspring Caused by Maternal Surgery during Mid-pregnancy

PONE-D-23-37577R1

Dear Dr. Luo,

We’re pleased to inform you that your manuscript has been judged scientifically suitable for publication and will be formally accepted for publication once it meets all outstanding technical requirements.

Kind regards,

Uwe Rudolph

Academic Editor

PLOS ONE

Additional Editor Comments (optional):

Reviewers' comments:

Reviewer's Responses to Questions

**Comments to the Author**

1. If the authors have adequately addressed your comments raised in a previous round of review and you feel that this manuscript is now acceptable for publication, you may indicate that here to bypass the “Comments to the Author” section, enter your conflict of interest statement in the “Confidential to Editor” section, and submit your "Accept" recommendation.

Reviewer #1: All comments have been addressed

Reviewer #2: All comments have been addressed

2. Is the manuscript technically sound, and do the data support the conclusions?

Reviewer #1: Yes

Reviewer #2: Yes

3. Has the statistical analysis been performed appropriately and rigorously? 

Reviewer #1: Yes

Reviewer #2: Yes

4. Have the authors made all data underlying the findings in their manuscript fully available?

Reviewer #1: Yes

Reviewer #2: Yes

5. Is the manuscript presented in an intelligible fashion and written in standard English?

Reviewer #1: Yes

Reviewer #2: Yes

6. Review Comments to the Author

Reviewer #1: (No Response)

Reviewer #2: This is a good manuscript demonstrating that treatment with the class 1 and 2b HDAC inhibitor SAHA can rescue cognitive deficits and negative changes in synapse-associated proteins induced by maternal surgery under propofol anesthesia in offspring.

The authors have revised the manuscript well as per the reviewers’ suggestions. The paper is good to go.

7. PLOS authors have the option to publish the peer review history of their article (what does this mean?). If published, this will include your full peer review and any attached files.

Reviewer #1: No

Reviewer #2: No

---

## [Editor Report · Acceptance letter]

17 Mar 2024

PONE-D-23-37577R1 

PLOS ONE

Dear Dr. Luo, 

I'm pleased to inform you that your manuscript has been deemed suitable for publication in PLOS ONE. Congratulations! Your manuscript is now being handed over to our production team.

Kind regards, 

on behalf of

Dr. Uwe Rudolph 

Academic Editor

PLOS ONE